# Immune Therapies for Myelodysplastic Syndromes and Acute Myeloid Leukemia

**DOI:** 10.3390/cancers13195026

**Published:** 2021-10-08

**Authors:** Sargam Kapoor, Grace Champion, Aparna Basu, Anu Mariampillai, Matthew J. Olnes

**Affiliations:** 1Hematology and Medical Oncology, Alaska Native Tribal Health Consortium, 3900 Ambassador Dr., Anchorage, AK 99508, USA; skapoor@anthc.org (S.K.); abasu@anthc.org (A.B.); aimariampillai@anthc.org (A.M.); 2School of Medicine, University of Washington, 1959 NE Pacific St., Seattle, WA 98195, USA; gchamp@uw.edu; 3WWAMI School of Medical Education, University of Alaska Anchorage, 3211 Providence Drive, Anchorage, AK 99508, USA

**Keywords:** acute myeloid leukemia, myelodysplastic syndromes, immune therapy, monoclonal antibody, antibody drug conjugate, CAR T-cells, vaccine therapy

## Abstract

**Simple Summary:**

Myelodysplastic syndromes (MDS) and acute myeloid leukemia (AML) are bone marrow cancers that have a poor prognosis. They have traditionally been treated with chemotherapy with limited success, and stem cell transplant, which many patients are too frail to receive. Perturbations in the immune system play a role in the development of MDS and AML. Recently, there has been progress in understanding how the immune system can be harnessed to treat these cancers. Effective immune therapies for MDS and AML have been developed at a rapid pace, including novel immune checkpoint inhibitor antibodies, bispecific T-cell-engaging antibodies, antibody drug conjugates, vaccine therapies, and cellular therapeutics including chimeric antigen receptor T-cells and natural killer cells. This review article highlights advances in this field for the practicing clinician to aid in understanding new developments for treating patients with MDS and AML.

**Abstract:**

Myelodysplastic syndromes (MDS) and acute myeloid leukemia (AML) are hematologic malignancies arising from the bone marrow. Despite recent advances in treating these diseases, patients with higher-risk MDS and AML continue to have a poor prognosis with limited survival. It has long been recognized that there is an immune component to the pathogenesis of MDS and AML, but until recently, immune therapies have played a limited role in treating these diseases. Immune suppressive therapy exhibits durable clinical responses in selected patients with MDS, but the question of which patients are most suitable for this treatment remains unclear. Over the past decade, there has been remarkable progress in identifying genomic features of MDS and AML, which has led to an improved discernment of the molecular pathogenesis of these diseases. An improved understanding of immune and inflammatory molecular mechanisms of MDS and AML have also recently revealed novel therapeutic targets. Emerging treatments for MDS and AML include monoclonal antibodies such as immune checkpoint inhibitors, bispecific T-cell-engaging antibodies, antibody drug conjugates, vaccine therapies, and cellular therapeutics including chimeric antigen receptor T-cells and NK cells. In this review, we provide an overview of the current understanding of immune dysregulation in MDS and AML and an update on novel immune therapies for these bone marrow malignancies.

## 1. Introduction

Myelodysplastic syndromes (MDS) are a clonal malignancy of the bone marrow, which presents in a heterogeneous manner with peripheral blood cytopenias and a potential for progression to acute myeloid leukemia (AML). The World Health Organization (WHO) classifies MDS into eight discrete categories according to blood and bone marrow morphology, the number of cell lineages involved, cytogenetic abnormalities, and bone marrow blast percentage, with up to 19% myeloblasts in the blood and/or marrow representing MDS with excess blasts and a level of 20% or higher defining AML (doi: 10.1182/blood-2016-03-643544) [1]. The most common WHO subtypes are MDS with single lineage dysplasia, ringed sideroblasts, multilineage dysplasia, excess blasts, and isolated deletion of chromosome 5q [1]. The various WHO subtypes are associated with a particular response to selected therapies, including lenalidomide for MDS with an isolated deletion 5q, and luspatercept for patients with ringed sideroblasts. The risk of death from MDS is due to complications from severe cytopenias and transformation to AML. MDS patient prognosis is further refined using prognostic scores, with the most widely utilized scale being the International Prognostic Scoring Scale-Revised (IPSS-R), which categorizes patients into risk groups ranging from very low to very high, with each risk category based on degree of cytopenias, bone marrow blast percentage, and cytogenetic abnormalities [2]. 

Over the past decade, there have been significant advances in understanding the molecular features of MDS and AML, which has enabled an improved refinement of disease classification, prognostic information, and novel therapeutic targets. The current WHO classification of AML recognizes the importance of discrete gene mutations and recurring cytogenetic abnormalities associated with prognosis [1]. The discovery of AML-associated gene mutations has led to novel therapeutics targeting FLT-3, IDH1, and IDH2, with other drugs on the horizon [3,4,5]. However, despite these advancements in understanding the molecular pathogenesis of AML, approximately two-thirds of patients will succumb from their disease within five years [6]. 

It is well recognized that there is an immune component to the pathogenesis of MDS and AML, but until recently, immune therapies have played a limited role in the treatment of these diseases. Insights into the immunobiology of MDS and AML have yielded emerging therapies, including novel monoclonal antibodies engineered to enhance antibody-dependent cytotoxicity and disrupt immune checkpoint interactions, bispecific and trispecific T-cell-engaging antibodies, antibody drug conjugates, vaccine therapies, and cellular therapeutics, including chimeric antigen receptor T-cells and NK cells. These emerging immune therapies have the potential to transform current treatment paradigms. In this review, we provide an update on current knowledge of immune dysregulation in MDS and AML and we provide an overview of novel immune-modulating therapies for these life-threatening bone marrow diseases, with particular emphasis on emerging treatment approaches.

## 2. Immune Dysregulation in MDS and AML

A robust body of knowledge supports the concept that immune dysregulation involving the full spectrum of innate and adaptive immunity contributes to the pathogenesis of MDS and AML [7,8,9,10,11,12]. It was appreciated as early as 1986 that MDS is more prevalent in patients with autoimmune and inflammatory conditions such as rheumatoid arthritis, vasculitis, thyroiditis, mixed connective tissue diseases, and polymyalgia rheumatica [13,14]. More recently, several cohort and registry studies have documented this association, with as many as 33% of patients with MDS having concomitant autoimmune diseases [15,16]. A population-based registration study in Sweden including 10,881 AML and MDS patients and 42,878 matched controls reported that a prior history of infectious disease was associated with an increased risk of MDS and AML, and a history of autoimmune disease was associated with 1.7- and 2.1-fold increased risk of developing AML and MDS, respectively [17]. This raises the possibility that prolonged immune stimulation may promote the development of MDS and AML [17]. A direct genetic link between autoinflammation and MDS/AML was discovered in the recently described VEXAS syndrome [18,19]. VEXAS syndrome is an adult-onset autoimmune systemic inflammatory syndrome including relapsing polychrondritis, Sweet’s syndrome, polyarteritis nodosa, or giant cell arteritis associated with myeloid malignancies, including MDS and AML [18,19]. VEXAS syndrome is caused by acquired somatic mutations in UBA1, the gene encoding ubiquitylation enzyme E1 in myeloid progenitor cells [18,19]. Treatment for VEXAS syndrome has yet to be clearly defined, but hypomethylating agents, janus kinase inhibitors, and allogeneic HSCT are being explored [20].

Patients with MDS exhibit increased levels of inflammatory cytokines, including TNF-α, IFN-γ, IL-6, IL-7, IL-8, and CXCL-10 expressed in bone marrow progenitor cells, macrophages, and stromal cells (Figure 1) [21,22,23,24,25]. Increased levels of CXCL10, IL-6, and IL-7 are associated with shorter MDS survival compared to patients with normal levels of these cytokines irrespective of age, transfusion burden, degree of cytopenias, and IPSS score [25]. TNF-α and IFN-γ induce apoptosis in CD34+ progenitor cells in MDS patients through transcriptional upregulation of programmed-cell-death-related genes TRAIL, Fas, and CASP8 [26,27]. 

Cell death within the bone marrow mediated through pyroptosis has been recently identified as a significant contributor to MDS pathogenesis (Figure 1) [12]. Pyroptosis is an inflammatory process of cell death mediated by caspase 1 that is distinct from apoptosis and is elicited by pathologic stimuli such as heart disease, stroke, and malignancies [28]. It is initiated by up-regulation of “danger signals” or danger-associated molecular patterns (DAMPs), which form the inflammasome, a multi-protein complex that activates caspase 1, leading to pores in the target cell membrane that trigger ion fluxes and ultimately cell death [28]. DAMP proteins include the innate immune signaling molecules toll-like receptors (TLRs) and other pattern recognition receptors (PRRs) such as the myeloid trans-membrane receptor CD33, which activates cells and stimulates the production of pro-inflammatory cytokines such as TNF-α, IFN-β, IL-6, and IL-8 [12,28]. Nod-like receptors, including nucleotide-binding oligomerization domain-containing protein 1 (NOD1) and NOD2, transduce signals from DAMPs in the target cell cytosol, stimulating additional inflammatory cytokines and activating caspase 1, which leads to cell death through pyroptosis [5]. 

Recent studies implicate dysregulated innate immune signaling in MDS, including overexpression of TLRs, IL-8 and its receptor CXCR2, and IL-1 downstream effector signaling pathways (Figure 1) [11,12,29]. TLR-4 signaling is increased in MDS CD34+ stem cells, and its expression correlates with levels of apoptosis [30]. Gene expression profiling studies also demonstrate that IFN-γ and chemokine signaling, as well as apoptosis-associated pathways, are overexpressed in CD34+ stem cells of patients with early MDS [31], and that activating mutations in TLR2 which upregulate IL-8 expression are present in stem cells of MDS patients [32]. 

Genomic interrogation studies have revealed further mechanistic details of inflammasome signaling in MDS, including downstream effectors of TLR signaling [11,29]. Deletion 5q (del(5q)) MDS is a well-studied model in this regard. Del(5q) MDS is characterized by haploinsufficiency micro RNAs miR-145 and miR-146a, which stimulate overexpression of downstream effects of TLR signaling, including TRAF-interacting protein-6, and interleukin-receptor-associated kinase (IRAK) proteins, which in turn up-regulate NF-κB [11,12,33]. Pharmacologic inhibition of IRAK-1 and -4 induces selective apoptosis of MDS CD34+ stem cells through a TRAF6/NF-κB-dependent mechanism without affecting normal stem cells [34]. Ligands for the TLR4 and CD33 receptors include the soluble inflammatory proteins S100A8 and S100A9, which bind their cognate receptors to trigger pyroptosis, as well as expansion of myeloid-derived suppressor cells (MDSCs) in the marrow, which suppress hematopoiesis through increased production of inflammatory cytokines IL-10, TGF-β, and granzyme-B [35]. Levels of S100A8 and inflammasome components are elevated in MDS patients [36,37], and murine studies demonstrate a critical role of alarmins in mediating pyroptosis in the MDS marrow [37]. Inflammasome signaling in the MDS marrow is amplified by mutations in genes encoding DNA (cytosine-5) methyltransferase 3A (DNMT3A) and Ten-eleven translocation 2 (TET2), which diminish their ability to quell inflammation through increased production of interferon-α and β, and upregulation of IL-6 through epigenetic regulation in dendritic cells and macrophages [38,39]. The MDS marrow microenvironment also favors enhanced innate immune signaling through downregulation of the epigenetic regulatory protein EZH2 [40] and upregulation of KDMGB, a protein that modulates the expression of innate immune response genes [41]. 

**Figure 1 cancers-13-05026-f001:**
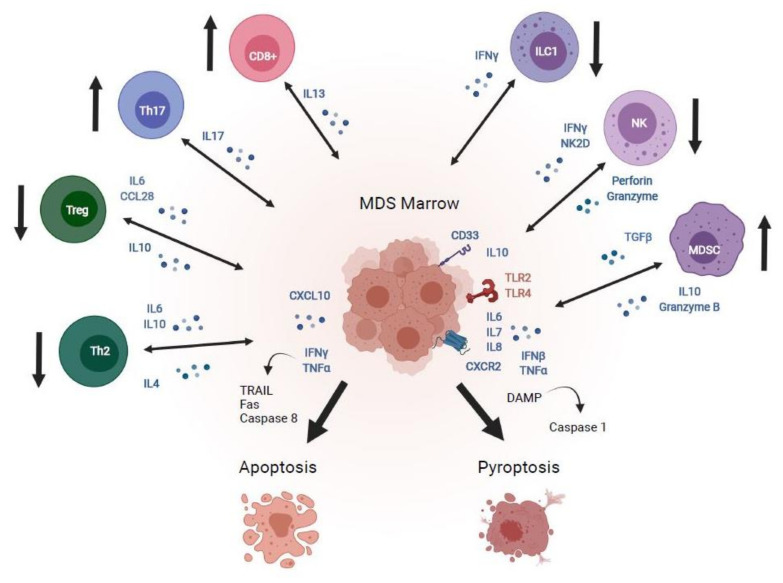
Mechanisms of immune dysregulation in MDS. The MDS bone marrow microenvironment exhibits perturbations in adaptive immune effector cells, including increased CD8+ cytotoxic T-cells and Th17 cells, which promote autoimmunity, as well as decreased Treg and Th2 cells, which dampen autoimmunity, favoring upregulation of IFNγ and TNFα, which induce apoptosis mediated by TRAIL, fas, and caspase-8. Innate immune derangements in the MDS marrow include decreased type 1 innate lymphoid cells (ILC1) and NK cells and increased myeloid derived suppressor cells, which secrete IL-10 and granzyme B, and promote signaling through CD33, toll-like receptors, and CXCR2 to enhance danger-associated molecular pattern stimulation of caspase-1, which promotes cell death via pyroptosis. Image created with BioRender.com.

Innate lymphoid cell (ILC) dysfunction has been observed in human malignancies including AML [42,43]. ILCs are innate immune cells classified into three groups based on cytokine production, cell surface receptors, transcription factor expression profiles, and shared effector functions [44]. Group 1 ILCs express the master transcription factor T-BET, which regulates expression of type 1 cytokines such as IFNγ, and includes NK cells and type 1 ILCs that react to intracellular pathogens and tumors [44]. Natural killer (NK) cell function is impaired in higher-risk MDS, with some patients exhibiting a quantitative deficiency in NK cell numbers, and other patients having normal cell numbers but deficiencies in levels of perforin and granzyme B and ineffective cell killing [45,46]. A reduced number of NK cells was associated with poor prognosis in high-risk MDS patients [46], suggesting that dysregulation of immune surveillance may contribute to disease pathogenesis. Group 2 ILCs express the transcription factor GATA3, secrete the type 2 cytokines IL-4, IL-5, and IL-3, and react to large extracellular pathogens and antigens; while group 3 ILCs express the transcription factor ROR γτ, produce IL-17 and IL-22, and react to extracellular microbes such as bacteria and fungi [44]. Multiple recent studies demonstrate dysregulation of ILC subsets in AML patients, including a hypo-mature NK cell phenotype, impaired NK cytotoxicity and decreased IFNγ secretion, decreased expression of the NK receptor activating surface marker NKG2D, and hypo-functional type 1 ILCs [43,47,48,49]. 

AML blasts evade immune surveillance by disrupting the immune microenvironment through multiple mechanisms, including downregulation of human leukocyte antigen class I and II presentation and upregulation of the immune checkpoint inhibitors programmed death ligand-1 (PD-L1), T-cells immunoglobulin-mucin 3 (TIM-3), and C-lectin-like inhibitory receptor-4 (CTLA-4) [49]. AML blasts also evade immune attack through immune editing, a mechanism by which AML cells function like myeloid-derived suppressor cells by secreting arginase-1, indoleamine 2–3 dioxygenase, and inducible nitric oxide synthetase, which suppress T-cell function by inducing anergy, cell cycle arrest, and programmed cell death [50]. Constitutive expression of STAT3 is central to the mechanism of immune editing in AML blasts and may represent a potential target to enhance susceptibility to immunotherapies [50].

Dysregulation of the adaptive immune response is also a well-described feature of the MDS bone marrow in a pattern characteristic of autoimmune states (Figure 1). The MDS T-cell repertoire exhibits clonal expansion of auto-reactive CD8+ cytotoxic T-cells, which suppress colony growth of MDS erythroid and granulocytic progenitors with trisomy 8 [51,52]. Patients with low to intermediate-1 MDS demonstrate increased levels of CD4+ T-helper cells, which enhance autoimmunity, and a decreased level of regulatory T-cells (Tregs) and Th2 cells that inhibit autoimmune inflammation [23]. Furthermore, interleukin-17-secreting CD4 + T-cells (Th17) are overexpressed in low-risk MDS patients, which correlates with apoptosis in the bone marrow and lower hemoglobin levels [23,53]. Autoreactive oligoclonal T-cells diminish with recovery of hematopoiesis in MDS patients who respond to immunosuppression [51,54]. On the other hand, impaired adaptive immune surveillance occurs in MDS patients with higher risk IPSS scores and in those with AML, with higher levels of Tregs and reduced numbers of CD4 + CD8 + effector memory T-cells, which may contribute to disease progression [49,55,56]. Collectively, these studies indicate that both innate and adaptive immune responses play a prominent role in the pathogenesis of MDS and AML and reveal potential novel targets for immunotherapies (Figure 1). 

## 3. Immunosuppression Trials for MDS

The recognition of immune perturbations in MDS patients led to clinical investigations into the use of immunosuppression therapy (IST) in MDS, with more than 20 studies published to date [57,58]. Initial studies were conducted using rabbit and equine anti-thymocyte globulin (ATG) given as a single agent or combined with cyclosporine, with transfusion independence rates ranging from 0% to 64% [59,60]. The studies were markedly heterogeneous with respect to therapies administered, patient characteristics, and enrollment criteria. Single-institution experiences identified features that predict for response to IST in MDS patients, including age <60, low-risk IPSS score, trisomy 8 karyotype, expression of the class II histocompatibility antigen DR15, and a shorter duration of transfusion independence [57,61]. 

An international multicenter retrospective cohort study was recently conducted in an effort to better define response rates in 207 MDS patients treated with IST at 15 centers in the United States and Europe [62]. The majority (91.3%) of patients included in this study had low or intermediate-1 IPSS MDS, and the median age was 61 years. ATG-based combination regimens were administered to most (76%) patients, and among the 125 patients for which response rate data was captured, the overall response rate was 48.8% (95% CI, 38.8–57.9%), with transfusion independence occurring in 30% (95% CI, 22.3–39.5%) for a median duration of 19.9 months (95% CI, 12.8–27 months). Complete responses were observed in 11.2% of patients (95% CI, 6.5–18.4%). Response rates correlated with overall survival (OS), with patients achieving a hematologic response having a median OS that was not reached (95% CI, 52.1 months—not reached), whereas non-responders had a median OS of 27.7 months (95% CI, 22.8–49.1 months; *p* = 0.0009). A multivariate analysis was performed to assess predictors for response, including all of the variables identified in single institution studies, and none of them demonstrated statistical significance. Among predictors for transfusion independence after IST, only a bone marrow cellularity of less than 20% was a predictor for red cell transfusion independence (odds ratio 4.0; 95% CI, 1.2–13; *p* = 0.03) [62].

More recently, a systematic review and meta-analysis of 22 published prospective cohort studies and clinical trials evaluating the use of IST in 570 MDS patients was performed [58]. The overall response rate in this meta-analysis was 42.5% (95% CI, 36.1–49.2%) among the included studies, with a red cell transfusion independence rate of 33.4% (95% CI, 25.1–42.9%), and a complete remission rate of 12.5% (95% CI, 9.3–16.6%). The most commonly studied regimen was ATG with cyclosporine, and none of the aforementioned predictors for response were confirmed in this study [58]. The CD52-directed monoclonal antibody alemtuzumab was investigated in a single-institution phase 1/2 clinical trial enrolling 32 MDS patients [63]. Hematologic improvements or complete responses were observed in 68% of evaluable patients treated with alemtuzumab, with a median response duration of 30 months [64]. However, there is limited availability of alemtuzumab as a treatment for MDS patients in many countries. Collectively, these studies indicate that IST with ATG combination regimens is associated with hematologic responses and improved survival in approximately 50% of patients, and red cell transfusion independence in approximately a third of patients. The majority of patients achieving hematologic responses are those with low and intermediate-1 MDS, and those with a hypocellular bone marrow are more likely to achieve transfusion independence.

## 4. Monoclonal Antibody Therapy for MDS and AML

### 4.1. Anti CD-47 (Macrophage Checkpoint Inhibitor) Therapies

The treatment options for MDS are limited, with no new therapies coming to light for more than a decade. However, two new drugs showed promise in 2020, one of which, luspatercept, was approved by the FDA in April 2020 for transfusion-dependent anemia in very-low- to intermediate-risk MDS with >15% ringed sideroblasts or <5% ringed sideroblasts with an SF3B1 mutation. Another drug, magrolimab, is a first-in-class monoclonal antibody that inhibits CD47 interactions with the signal regulatory protein alpha (SIRPα) receptor on macrophages. Interruption of CD47 with SIRPα blocks the protective mechanism of cancer cells against macrophage phagocytosis and serves as a macrophage checkpoint inhibitor [65]. Magrolimab received FDA breakthrough designation for the treatment of newly diagnosed MDS in September 2020. This was based on positive results from a phase 1b study that studied magrolimab in combination with azacitidine for the upfront treatment of intermediate, high-risk, and very-high-risk MDS [66,67]. In vitro and early phase human studies of magrolimab raised concerns about platelet aggregation, increased erythrocyte clearance, red blood cell agglutination, and hemolysis [65,68]. However, in the aforementioned phase 1b study, this combination was well tolerated and there were no toxicity-related withdrawals from the study. In 13 evaluable MDS patients treated with magrolimab and azacitidine, all patients (100%) achieved an objective response, and 7 patients (54%) achieved a complete remission (CR) [67]. This drug is now being studied in the phase 3 ENHANCE trial in previously untreated higher-risk MDS (Table 1).

Magrolimab is currently being studied for several hematologic malignancies. Overexpression of CD47 on AML cells and its negative impact on prognosis [75,76] has formed the basis for its use in this challenging disease [77] (Figure 2). In December 2020, a phase 1b trial showed encouraging results for patients with untreated AML who are unfit for intense induction chemotherapy. Among 34 evaluable patients treated with magrolimab, 65% achieved an objective response, 44% achieved a CR, and 12% achieved a CR with incomplete count recovery (CRi). Remarkably, this combination appeared effective against the notorious TP53 mutant AML as well, with 71% of patients achieving an objective response, 48% achieving a CR, and 19% achieving a CRi in TP53 mutant AML patients. The median duration of response was 9.9 months for all patients, and median OS was 18.9 months (95% CI 4.34 months- NR) for TP53 wild-type patients, and 12.9 months (95% CI 6.24 months—NR) for TP53 mutant patients. The incidence of adverse effects reported was 15%, these being mostly cytopenias, liver function test abnormalities, and infusion-related reactions, with treatment discontinuation in 4.7% patients [66]. Magrolimab is under study in other early phase clinical trials. NCT02678338 (CAMELLIA) enrolled patients with R/R AML and high-risk MDS in the United Kingdom, (Table 1). The phase 1 results however reported a decrease in hemoglobin, an increase in transfusion requirements, and issues with ABO compatibility testing with this treatment [69].

Other anti-CD47 monoclonal antibodies are currently under investigation, including the humanized mAb CC-90002. Pre-clinical studies showed activity of CC-90002 in enabling macrophage-mediated killing of AML cells. However, a phase 1 multicenter clinical trial, NCT02641002, that enrolled patients with relapsed/refractory (R/R) AML and R/R high-risk MDS for treatment with this drug as monotherapy was terminated early due to lack of objective responses and development of anti-drug antibodies (Table 1) [70]. TTI-621 (SIRPαFc) is a soluble recombinant fusion protein developed by linking the N-terminal CD47-binding domain of SIRPα with the Fc domain of IgG1 that is being studied for patients with several R/R hematologic malignancies, including AML, as part of phase 1 trial NCT02663518 (Table 1). ALX148 is another fusion protein that blocks the CD47-SIRPα pathway. It is being studied in phase 1/2 clinical trial NCT04755244 in combination with venetoclax and azacitidine for previously untreated R/R AML patients who are unfit for intense induction chemotherapy (Table 1). While anti-CD47 monoclonal antibodies are an encouraging immune checkpoint treatment for MDS and AML on the horizon, use is restricted to clinical trial settings at this time.

**Figure 2 cancers-13-05026-f002:**
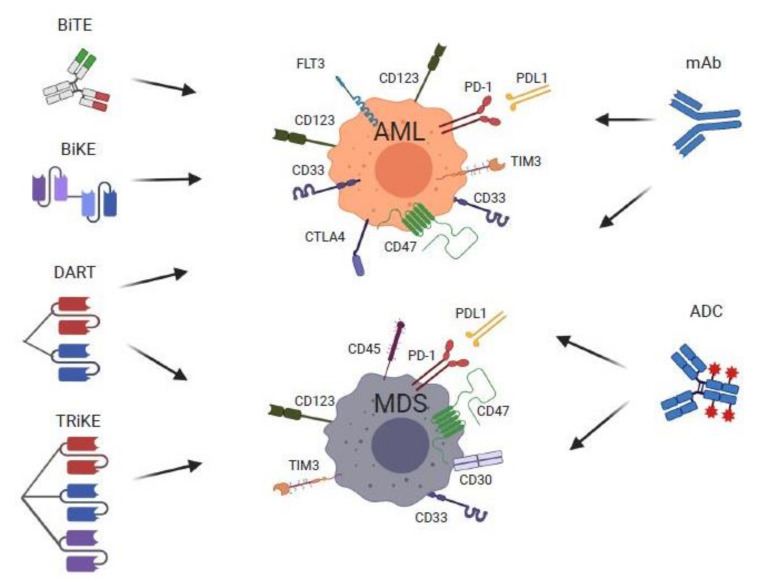
Monoclonal antibodies (mAb), antibody-drug conjugates (ADC), and novel engineered monoclonal antibody therapies for MDS and AML and their immune targets (**Middle panel**). Emerging mAb immunotherapies include antibodies directed against immune checkpoint molecules PD-1/PD-L1, CTLA-4, and TIM3, the macrophage mediated phagocytosis inhibitor CD47, and Fc-optimized mAbs targeting CD33 and CD123. Engineered mAb therapies (**Left panel**) have a basic common mechanism of action by forming bonds between a tumor-associated antigen and CD3, resulting in physical interaction between T-cells and leukemic cells. These molecules include bi-specific T-cell engagers (BiTEs)—fusion of two single-chain variable fragments (scFvs) of two unique antibodies (red and green) into a single peptide chain with affinity of one to T-cells via CD3 and the other to tumor molecules, dual-affinity re-targeting molecules (DARTs)—similar to the BiTE construct with addition of a disulfide linker for increased stability, bi- and tri-specific killer cell engager (BiKEs and TriKEs)—molecules consisting of a single-chain variable fragment (scFv) fused by a heavy and variable chain, that are linked to the scFV of one (BiKE) or two (TriKE) antibodies via CD16, a receptor on NK cells that upon stimulation activate various cytokines and induce a cytolytic response by targeting CD33, CD123, and FLT-3 on tumor cells. ADCs include compounds targeting CD33, CD123, CD45, and CD30. Image created with BioRender.com.

### 4.2. Immune Checkpoint Inhibitor Therapies

Immune checkpoint inhibitors have revolutionized the treatment landscape of solid tumor oncology over the last decade. Their basic principle is based on targeting immune checkpoints that cause the tumor cell’s escape from immune surveillance, including (i) CTLA-4 inhibitors, which prevent CTLA-4 on the tumor cell from binding to B7 on the T-cell/antigen-presenting cell that causes downregulation of T-cell activity and (ii) PD-1/PD-L1 inhibitors that prevent T-cell inactivation that is a result of programmed death-1 (PD-1) (T-cell) binding with programmed death –ligand 1 (PD-L1) (on tumor cell). T-cell immunoglobulin mucin-3 (TIM-3) and lymphocyte activation gene 3 (LAG-3) are more recent immune checkpoint targets being studied. TIM-3 is a glycoprotein on the surface of T-cells and innate immune cells, which downregulates Th1 immune responses. LAG-3 is a transmembrane receptor on activated T-cells, T-regs, and NK cells that suppresses T-cell and NK cell responses. Amongst hematologic malignancies, overexpression of PD-1 ligands by Reed–Sternberg cells in classic Hodgkin lymphoma formed the basis of study and eventually approval of these drugs for R/R disease. These agents are currently under study in MDS and AML either alone or in various combinations with conventional MDS/AML therapies. 

#### 4.2.1. Anti-CTLA-4 Therapies

Increased expression of B7 on antigen-presenting cells correlates with inferior outcomes in AML [78,79]. This formed the basis for studying CTLA-4 inhibitors in this disease to prevent the downregulation of immune defense against AML blasts. One such trial investigated ipilimumab, an anti-CTLA-4 antibody in several hematologic malignancies that relapsed in the post-allogenic HSCT setting. The hypothesis of this study was to prevent tumor evasion of the donor immune system and strengthen the graft vs. leukemia effect. Complete responses were noted in four extramedullary AML patients and one patient with secondary AML out of a total of 13 patients with R/R AML [71]. Adverse events included GVHD and immune-mediated pneumonitis and colitis. Another phase 1b study of ipilimumab in 29 relapsed/refractory high-risk MDS patients that progressed on hypomethylating agents showed marrow CR in only one patient (3.4%) but prolonged stable disease for more than 46 weeks in seven patients (24%), including three patients that had stable disease for over a year, and in five patients able to undergo an allogeneic HSCT without severe toxicities [80]. Interestingly, patients with higher expression of the inducible co-stimulator (ICOS) molecule had better outcomes. 

#### 4.2.2. Anti-PD-1 and Anti-TIM-3 Therapies

Increased expression of PD-1 on MDS and AML cells has been documented, especially in patients treated with hypomethylating agents [81,82]. However, PD-1 inhibition alone, while safe, did not seem to make any beneficial impact on disease outcomes [83]. This formed the basis of studies combining PD-1 inhibition with hypomethylating agents in R/R MDS or AML. One such phase 2 single-arm study enrolled 70 patients with R/R AML treated with azacitidine in combination with nivolumab given on days 1 and 14 every 4–6 weeks. They reported an overall response rate of 58% in the hypomethylating-agent-naïve arm and 22% in the hypomethylating-agent-pre-treated arm. Complete responses were observed in 15 patients (22%); one partial response, and seven had hematologic improvement for more than six months. Six of 70 patients had stable disease for more than six months. The combination was effective and safe in patients that were hypomethylating-agent-naïve, in the salvage setting, or in patients with increased CD3+ cells in the bone marrow [72]. An additional cohort of R/R AML patients treated with azacitadine, nivolumab and ipilumimab demonstrated a median OS of 7.6 months, with OS in the azacitidine/nivolumab arm being 5.9 months, and 4.4 months in the hypomethylating-agent-treated control arm [84].

Another trial studied nivolumab in high-risk AML patients in remission but ineligible for allogeneic HSCT, with high risk defined as secondary AML in first CR, high-risk cytogenetics, FLT-3 ITD mutation, presence of minimal residual disease, or second CR or more [73]. This was based on the hypothesis that immune checkpoint inhibitors may be able to generate an anti-tumor response akin to the graft vs leukemia effect in the allo-HSCT setting. Only 2/9 MRD-positive patients cleared the MRD on nivolumab. Over a median follow-up of 30.4 months, the relapse-free survival was 57.1% and median relapse-free survival was 8.48 months, which is similar to historical data with observation alone [73]. While safe and feasible, nivolumab monotherapy did not seem promising in this setting. With the emergence of oral azacitidine in the maintenance setting, attention has now shifted to combination therapies post-remission. Of note, the combination of durvalumab and azacitidine had previously failed to improve PFS vs. azacitidine alone in older patients with high-risk MDS and AML [85]. Pembrolizumab is also being studied in MDS and AML, both in the frontline setting in fit adults in combination with 7 + 3 induction chemotherapy and in unfit adults in combination with azacitidine and venetoclax. Pembrolizumab is also being studied in R/R MDS after failure on hypomethylating agents in combination with entinostat (Table 1). TIM-3 is not only expressed on immune cells as mentioned above, but also, expression is high in AML stem cells, likely causing downregulation of effector T-cell function [72]. Moreover, TIM-3 is not present on normal hematopoietic stem cells. Anti-TIM-3 antibodies are thus under study in myeloid malignancies, with preliminary results from a phase 1 trial studying sabatolimab (MBG453) with hypomethylating agents demonstrating an ORR of 41% in 34 AML patients, and an ORR of 62% in 35 MDS patients [74] (Table 1). There are several TIM-3 antibodies being investigated in a variety of malignancies. Upregulation of TIM-3 in tumor cells that have become resistant to PD-1/PD-L1 blockade might be overcome by TIM-3 antibodies either administered alone or in combination with other immune checkpoint inhibitors [86]. In short, immune checkpoint inhibitor therapies are being explored as part of combination therapy for anti-leukemic effects in clinical trial settings at this time.

### 4.3. Antibody-Dependent Cellular Cytotoxicity (Anti-CD33 and CD123 Fc-Optimized Antibodies)

Fc-optimized monoclonal antibodies are engineered to enhance recruitment of FcγRIIIa-positive effector cells in an effort to improve antibody-dependent cellular cytotoxicity (ADCC). In addition to targeting the AML antigens, Fc-optimized antibodies increase NK cell activation to enhance ADCC. This is in contrast to antibody drug conjugates, which are combined with a cytotoxic agent but do not augment the immune response [77]. Fc-optimized antibodies have therefore been developed against the widely investigated AML antigens CD33 and CD123. CD33 is also highly expressed in MDS suppressor cell populations [86].

Although NK cell activity is impaired in AML [43,47,48,49,87], preclinical data has shown enhanced response and NKG2D ligand upregulation on combining decitabine with an anti-CD33 Fc-optimized antibody BI 836858 [88]. This has formed the basis for clinical trials using BI 836858 in AML (Table 1). The failure of CSL-360, a naked monoclonal antibody against CD123 (IL-3 signaling), triggered the need to develop an Fc-optimized anti-CD123 antibody that would induce NK cytotoxicity [89]. CSL362 served this purpose with a modified Fc domain increasing NK cell ADCC. Similar to BI 836858, pre-treatment with decitabine increased NK cell activity in response to CSL362 [90]. Notably, this drug was used in high-risk patients in CR in a phase I study, and 10/20 evaluable patients remained in CR at 6 months [91]. Further data are needed to determine if Fc-optimized CD123-directed antibodies will elicit toxicities such as capillary leak syndrome. There are ongoing clinical trials utilizing CSL362 in combination with decitabine in unfit patients with AML (Table 1). 

## 5. Bispecific T-Cell Engagers, Dual-Affinity Re-Targeting Molecules, Bi- and Tri-SPECIFIC Killer Cell Engager Therapies for MDS and AML

Bispecific T-cell engagers (BiTE) are a promising new area of research in MDS and AML. BiTEs are derivatives of two single-chain antibody fragments (scFv) from two different antibodies, each targeting components of the effector T-cells (the epsilon subunit of T-cells—CD3ε) to an associated cell surface antigen on malignant cells, thus facilitating close approximation and promoting T-cell-mediated killing of malignant cells (Table 2). Effective BiTE therapy requires a unique target that is specific and crucial to the survival of malignant cells while sparing normal hematopoietic cells. The cell surface proteins CD33 and CD123 have been targets of intense investigation, as these are often expressed on the surface of aberrant myeloid cells and in over 90% of AML (Figure 2). 

AMG330 and AMG673 (half-life extended version) are two BiTE cell constructs developed with specificity for CD33 and CD3 epsilon. In a recent human phase 1 dose escalation trial, AMG330 showed antitumor activity in R/R AML, with 11.4% patients achieving CR/CRi. However, a majority of patients (87%) discontinued treatment due to progression of disease, while 73% experienced serious adverse events, including cytokine release syndrome (CRS) and cytopenias related to treatment [92]. A recent study update showed safety in dosing up to 720 ug/day, with 8/42 (19%) patients responding [92]. AMG330 has also shown synergy with PD-1/PD-L1 axis blockade (Table 2) [93]. Phase 1 studies of AMG 673 showed a 44% (12/27) decrease in BM blasts, with 22% (6/27) achieving >50% decrease in blasts compared to baseline. One patient achieved CRi with 85% reduction in BM blasts. Pharmacokinetic and pharmacodynamics were consistent with biological activity of the drug [93,94].

**Table 2 cancers-13-05026-t002:** Clinical trials of various antibody constructs (BiTE and TriKE) in therapy of MDS and AML.

Study	Trial Identification	Mechanism	Study Population	Study Status	Results	MRD Status	Adverse Effects	Ref
Bispecific T-Cell Engagers (BiTE)								
Phase 1 dose escalation (AMG 330)	NCT02520427	Bispecific mAb w/ specificity to CD33 and CD3ε	R/R AML	Recruiting	CR/CRi: 16%	N/A	CRS: 67%	[92]
Phase 1b dose escalation, AMG 330+ pembrolizumab	NCT04478695	Bispecific mAb w/ specificity to CD33 and CD3ε + PD-L inhibitor	R/R AML	Terminated	N/A	N/A	N/A	N/A
Phase 1 dose escalation (AMG 427)	NCT03541369	Bispecific mAb w/ specificity to CD3ε and FLT3 scFv	R/R AML	Active, Recruiting	N/A	N/A	N/A	[95]
Phase 1 dose escalation (AMV 564)	NCT03144245	Bispecific mAb w/ specificity to CD33 and CD3ε- tetravalent	R/R AML	Active, Not Recruiting	Reductions in BM blasts ranging from 13 to 38% in 6 of 9 evaluable patients	N/A	Febrile neutropenia in 25%	[90]
Phase 1 dose escalation Vibecotamab (XmAb14045)	NCT02730312	Bispecific mAb w/ specificity to CD123/CD3ε	R/R AML	Active, Recruiting	ORR: CR/CRi/MLFS: 14% SD:71% Low disease burden: RR 26%	N/A	CRS: 58%	[96]
Phase 1 dose escalation (AMG 673)	NCT03224819	Half-life extended bispecific T-cell engager—HLE BiTE to CD33 and CD3	R/R AMl	Active, Not Recruiting	ORR 44% decrease in BM blasts, 22% with >50% reduction and 3% with >85% reduction in BM blasts	N/A	CRS: 50%,	[93]
Dual Affinity Re-Targeting Molecules DART (BiTE)								
Phase 1/2 Flotetuzumab (MGD006)	NCT02152956	Bispecific mAb w/ specificity to CD123/CD3ε	PIF or R/R AML or Intermediate-2/High Risk MDS	Active, Recruiting	CR/CRh: 24% Median OS (in patients that achieved CR/CRh): 10.2 m	NA	IRR/CRS in PIF/ER population (n = 30): 100% (all grades), Gr > 3: 3.3%	[97]
Tri-Specific Killer Engager (TriKE)								
Phase 1—GTB-3550 TriKE	NCT03214666	Tri-Specific Killer Engager-CD16/IL-15/CD33	PIF/ R/R AML, High risk MDS	Active, recruiting	4 pts enrolled, 3 completed, 2 w/SD, 1 w/POD.	NA	No toxicity reported.	[98]

Abbreviations: R/R, relapsed refractory; CR, complete remission; CRi, complete remission with incomplete count recovery; CRh, complete remission with partial hematologic recovery; MLFS, morphologic leukemia-free state; CRS, cytokine release syndrome; BM, bone marrow; LFT, liver function tests; POD, progression of disease; RR, response rate; PIF, primary induction failure; ER, early relapse; MDS, myelodysplasia; IRR, infusion-related reactions.

More recently, BiTE constructs targeting the Fms-related receptor tyrosine kinase 3 (FLT3) protein have also been developed. Early work evaluated the FLT3-directed BiTE construct AMG 427 using mouse xenograft models, and AMG427 treatment in an animal model showed bone marrow FLT3 transcript reduction by 85–92% compared to pre-treatment levels [95]. Patients treated with weekly doses of AMG 427 in a clinical trial achieved an improvement in median survival from 36 to over 108 days (n = 10, *p* < 0.001). AMV564, another BiTE construct with specificity to CD33 on target cells and CD3 on effector T-cells is currently undergoing phase 1 dose escalation study in R/R AML patients that have undergone at least 1–2 lines of prior induction regimens, with preliminary data showing a 49% reduction in bone marrow blasts [96]. Vibecotamab (XmAb1404) is a BiTE construct with specificity to CD123 and CD3 that was evaluated in a phase 1 dose escalation study [99]. Treatment was administered weekly in 28-day cycles. Dosages ranged from 0.003 to 12.0 ug/kg. Results from 104 heavily pretreated patients with R/R AML, B-cell ALL, and CML, including some who had prior allogeneic HSCT, showed ORR of 14% of patients treated at a dose of 0.75 ug/kg, with 4% of patients achieving a CR, and three with CRi. Stable disease was noted in 71% of patients (Table 2). The most common toxicity was CRS, seen in 58% of patients, 85% of which were grade 1–2, and 15% grade 3. Interestingly, responders were found to have a lower disease burden and specific T-cell subtypes, suggesting further molecular characterization would be of benefit with this therapy [99].

Dual-affinity re-targeting (DART) molecules are antibody constructs that consist of two scFv fragments with different VH domains, which allows binding to two unique cell surface molecules simultaneously (Figure 2). This allows recruitment of T-cells or NK cells without the limitation inherent to BiTEs, resulting in improved stabilization of the molecule. Flotetuzumab (MGD006) is a bispecific DART molecule engineered for binding of CD3ε and CD123 of AML cells. A phase 1/2 study evaluated the use of Flotetuzumab dosed at 500 ng/kg per day administered as continuous infusion in R/R AML patients with primary induction failure (PIF) and early relapse (ER) with a median of four prior lines of therapy (range 1–9) (Table 2) [97]. Among 88 patients enrolled, the ORR was 13.6%, of which 10 (11.7%) were CR/CRi. Decreases in circulating blasts were seen in all dosing cohorts. Importantly, subgroup analysis showed increased Flotetuzumab activity (>50% blast reduction) in the PIF/ER group (43%) compared to late relapse (14%) patients, suggesting increased sensitization of leukemic cells with infiltrated/inflamed tumor micro-environment to Flotetuzumab. In the PIF/ER population, combined CR/CR with partial hematologic recovery was 26.7%, with an overall response rate of 30%, a median OS of 10.2 months, and 6- and 12-month survival rates of 75% and 50%, respectively (Table 2). The most frequent adverse reactions in the trial were infusion-related reactions and CRS [97].

An alternative to T-cell targeting is to recruit the anti-tumor function of NK cells, which confer enhanced immune recognition to tumor cells that are classically less immunogenic, and thus unrecognized by T-cells. Bispecific killer cell engagers, or BiKEs, are an alternative to BiTEs in that they target tumor antigens along with the NK marker CD16, resulting in persistent NK cell activation and cytotoxicity towards tumor cells. NK cells mediate cytotoxic effects via degranulation, secretion of potent cytokines, and induction of apoptosis via surface ligands. By targeting CD33 found on leukemic cells with CD16 on NK cells, BiKEs deliver a more potent mechanism of tumor destruction [100]. TriKEs build upon BiKES by the incorporation of IL-15, a potent stimulus for activating and expanding NK cells, thus potentially resulting in more sustained in vivo presence of expanded NK cells (Figure 2) [101]. TriKEs are constructed by fusion of three scFv segments, resulting in a single-chain triple body. TriKEs have yielded encouraging anti-leukemic results in animal models [102]. GTB-3550 is a single-chain tri-specific scFV recombinant fusion protein conjugate composed of the variable regions of the heavy and light chains of anti-CD16, anti-CD33 antibodies, and a modified form of IL-15. Preliminary data from an ongoing phase 1/2 expansion trial with GTB-3550 evaluating four enrolled patients demonstrated no clinically significant targeted toxicity at 5 and 10 mcg/kg dosing; two patients had stable disease, one showing progression without any reports of CRS or IRR (Table 2) [98]. Correlative studies showed increased NK cell activation during treatment, starting at day three, with maximal activation at day 9 and sustained activity at day 15 and 22 [98]. Additional studies of BiTEs, DARTs, BiKEs, and TriKEs for MDS and AML are warranted.

## 6. Antibody Drug Conjugate Therapies for MDS and AML

### 6.1. Anti-CD33 ADC Therapies

#### 6.1.1. Gemtuzumab Ozogamicin

Antibody-drug conjugates (ADCs) are mAbs directed toward tumor-associated antigens, to which highly potent cytotoxic agents are attached using chemical linkers. Multiple AML tumor antigens have been identified as targets for ADCs, and a variety of ADCs have been studied in patients with AML and MDS (Figure 2) [103]. CD33 is a myeloid cell surface antigen that is expressed on blast cells in acute myeloid leukemia. Gemtuzumab ozogamicin (GO) is a mAb directed against CD33 linked to a cytotoxic derivative of calicheamicin that was the first ADC approved for use in humans. Following a dose-escalation study, GO was initially investigated in three open-label multicenter single-arm studies for patients with CD33+ AML in first relapse administered at 9 mg/m^2^ in two doses separated by two weeks [104]. The ORR were 34% and 26% in patients <60 years of age and those ≥60 years old, respectively. The relapse-free survival (RFS) rate of all who attained a remission was 6.8 months, with 7.2 months for those achieving a CR, and 4.4 months for the complete response with incomplete platelet recovery (CRps). However, patients over 60 years of age had a much shorter RFS of 2.3 months versus 17 months for patients <60 years of age. Remission rates were 28% and 32% for patients with durations of first remission <1 year and ≥1 year, respectively. Side effects of GO included treatment-related infusion reactions such as fever, chills, hypotension, hematological toxicity, and infection. Based on this data, GO was approved by the FDA in 2000 for patients over 60 years with relapsed AML who were not candidates for standard cytotoxic chemotherapy (Table 3) [104].

GO was then studied in the de novo AML setting in the SWOG S0106 study, in which patients received induction therapy with daunorubicin and ara C (DA) with and without GO [109]. This study demonstrated no overall improvement in survival and increased treatment-related mortality in the patients treated with GO, which led to its withdrawal from the commercial market in October 2010 [100]. A French open-label multicenter trial, ALFA-0701, randomized 271 patients aged 50 to 70 with untreated de novo AML 1:1 with GO dosed at 3 mg/m^2^ on days 1, 4, and 7, plus DA [110]. Complete response with or without incomplete platelet recovery to induction was 104 (75%) in the control group and 113 (81%) in the GO group (odds ratio 1·46, 95% CI 0.20–2.59; *p* = 0.25). Event-free survival (EFS) was 17.1% (10.8–27.1) at two years in the control group, versus 40.8% (32.8–50.8) in the GO-treated group (hazard ratio 0.58, 0.43–0.78; *p* = 0.0003), and OS was 41.9% (33.1–53.1) versus 53.2% (44.6–63.5), respectively (0.69, 0.49–0.98; *p* = 0.0368), and RFS 22.7% (14.5–35.7) versus 50.3% (41.0–61.6), respectively (0.52, 0.36–0.75; *p* = 0.0003) [110]. A subsequent updated analysis of the same trial showed no difference in ORR or OS but a continued difference in EFS, with a median EFS of 17.3 months vs 9.5 months in the GO vs non-GO arms (HR 0.56, 0.42–0.76, *p* < 0.001). Subgroup analyses stratified for EFS and OS by age, sex, ECOG, CD33 expression, and cytogenetic risk categories demonstrated benefit for all patients except those with AML with adverse cytogenetics [111].

The efficacy of GO in frontline AML was confirmed in a meta-analysis of five trials, which included a total of 3325 patients [112]. Remission rates were not increased, but by reducing the risk of relapse, the OS at five years was improved irrespective of patient age (30.7% vs 34.6%; HR 0.90 (95% CI 0.82–0.98), *p* = 0.01). The survival benefit was particularly clear in those with favorable cytogenetics (55.2% vs 76.3%; HR0.47 (0.31–0.73), *p* = 0.0005), but also observed in intermediate-risk patients (34.1% vs 39.4%; HR 0.84 (0.75–0.95), *p* = 0.007). Patients with adverse karyotype did not benefit overall or within any trial [112].

GO was also studied as monotherapy versus best supportive care (BSC) as first-line therapy in 237 older patients with AML unsuitable for intensive chemotherapy [113]. Those with no evidence of disease progression or significant toxicities after induction were eligible to receive continuation therapy with a single monthly dose of GO 2 mg/m2 every four weeks for up to eight courses. The median OS was 4.9 months (95% CI, 4.2 to 6.8 months) in the GO group, and 3.6 months (95% CI, 2.6 to 4.2 months) in the BSC group (hazard ratio, 0.69; 95% CI, 0.53 to 0.90; *p* = 0.005), and the 1-year OS rate was 24.3% with GO and 9.7% with BSC. The OS benefit with GO was consistent across most subgroups and was especially apparent in patients with high CD33 expression, in those with favorable/intermediate cytogenetic risk profile, and in women. Complete remission occurred in 30 of 111 (27%) GO-treated patients. The rates of serious AEs were similar in the two groups, and no excess mortality from AEs was observed with GO [113]. Based on this data, GO can be used as single agent or in combination with chemotherapy in both de-novo and relapse setting. 

#### 6.1.2. Other CD33-Directed ADCs: Vadastuximab Talirine, IMGN779 and AVE9633

Vadastuximab talirine (VT) is a CD33-directed antibody conjugated to pyrrolobenzodiazepine (PBD) dimers. A phase 1 study assessed safety, tolerability, and activity of vadastuximab talirine with hypomethylating agents [105]. No dose-limiting toxicities were reported. The majority of adverse events were a result of myelosuppression, with some causing therapy delays. The complete remission rate, including CRi, was 70% in 53 patients treated. Median relapse-free survival and overall survival were 7.7 and 11.3 months, respectively. Compared with historical data for HMA monotherapy, the combination of VT with HMAs produced a high remission rate but was accompanied by increased hematologic toxicity (Table 3) [105]. However, the phase 3 cascade trial evaluating VT in combination with HMA compared to hypomethylating agent alone in older patients with newly diagnosed AML was halted after reviewing unblinded data indicating a higher rate of death, including fatal infections in the VT-containing arm of the trial. 

The ADC IMGN779 comprises a humanized anti-CD33 mAb conjugated via a cleavable disulfide linker to DGN462, a DNA-alkylating agent. IMGN779 was found to be highly active in vitro against primary patient AML cells, with cells harboring FLT3-ITD mutations being more sensitive to IMGN779 inhibition compared with FLT3 WT AML samples [114]. IMGN779 showed manageable toxicity and encouraging anti-leukemic activity in a pilot phase 1 trial enrolling R/R AML patients (Table 3) [106].

AVE9633 is an anti-CD33-maytansine conjugate ADC studied in three phase 1 trials, enrolling a total of 54 patients with R/R AML [115]. Patients were infused AVE9633 on day one of a 21-day cycle (Day 1 study); day one and eight (Day 1/8 study); and day one, four, and seven (Day 1/4/7 study) of a 28-day cycle. One CR, one PR, and biological activity in five other patients were observed in this study. The Day 1 and Day 1/4/7 studies were discontinued early because of drug inactivity at doses significantly higher than CD33-saturating levels. Given the minimal clinical activity seen, there are no further studies being pursued [115].

### 6.2. CD123-Directed ADCs: IMGN632, SGN-CD123A, SL-101, and SL-401

The IL-3 receptor α chain CD123 is expressed on a majority of AML blasts and has been found to be expressed more in leukemic cells relative to normal hematopoietic stem cells, making it an attractive target for immune-modulating therapies. Several CD123-targeting ADCs have been developed. IMGN632 is a humanized anti-CD123 antibody G4723A linked to a recently reported DNA mono-alkylating payload of the indolinobenzodiazepine pseudodimer (IGN) class of cytotoxic compounds [107]. A phase 1 trial in relapsed and refractory AML demonstrated objective responses in 1/3 of patients [107]. SGN-CD123A is another ADC composed of a humanized anti-CD123 mAb with a PBD dimer attaching an engineered cysteine residue on each one of the two heavy chains via a protease-cleavable dipeptide linker. Preclinical studies showed that SGN-CD123A demonstrated anti-tumor activity against AML cell lines and primary samples from AML patients with or without adverse cytogenetic profiles or FLT3 mutations [116]. SGN-123A induced remission in AML xenograft models and significant growth delay in a multidrug-resistant xenograft, and it enhanced the activity of FLT3 inhibitor quizartinib in two FLT3 mutated xenograft models [116]. However, the first in-human phase 1 trial in AML patients (NCT02848248) was terminated in May 2018 because of safety concerns. SL-101 is a novel anti-CD123 antibody-conjugate comprised of anti-CD123 scFv fused to a truncated and optimized pseudomonas exotoxin lacking its native targeting domain. Preclinical studies have demonstrated SL-101′s cell-killing efficacy in AML cell lines and primary AML cells [117]. Phase 1 trials are currently in development. Finally, SL 401 (Tagraxofusp) is a CD123-directed cytotoxin consisting of recombinant human interleukin 3 fused to a truncated diphtheria toxin that is FDA approved for the treatment of blastic plasmacytoid dendritic cell neoplasm [118]. This compound was studied in a phase 1 trial enrolling de novo, R/R MDS, and AML patients, with one patient achieving a durable CR of 8 months, two patients exhibiting partial responses lasting one and three months, and three patients with minimal responses with clearance of peripheral blasts and marrow blast cytoreductions of 89%, 90%, and 93% (Table 3) [119]. SL 401 was also studied in a Phase 2 trial as consolidation therapy for patients with AML in remission with high relapse risk, including those with minimal residual disease (MRD) [120]. The safety profile was similar to previous trials [120]. Further studies are warranted to determine the efficacy of this approach. A phase 1 trial of SL-401 in combination with azacitidine or azacitidine/venetoclax in AML, high-risk MDS, and blastic plasmacytoid dendritic cell neoplasm is currently ongoing (NCT03113643).

### 6.3. Anti-CD45 and -CD30 Directed ADCs for MDS and AML

CD45 is widely expressed on AML blasts and is thought to play a role in AML development and maintenance [121]. This molecule has emerged as a target for radioimmunotherapy as part of the conditioning regimen prior to allogenic HSCT. One such ADC is the compound 131I-BC Ab, an anti-CD45 antibody conjugated to iodine 131, which is also known as Iomab-B [108]. 131I-BC Ab was studied in combination with a reduced-intensity conditioning regimen of fludarabine plus 2 grays total body irradiation, in 58 AML or high-risk MDS patients older than 50 years of age in a clinical trial to estimate maximum tolerated dose (MTD) [108]. Treatment produced a complete remission in all patients and engraftment by day 28 of transplant. Median OS and DFS among all 58 patients was 199 days and 159 days, respectively, and among the 21 patients treated at the MTD was 206 and 189 days, respectively. Adverse effects included infusion toxicities, chills, nausea, vomiting, respiratory symptoms such as throat or chest tightness, and hypotension [108]. A phase 3 SIERRA trial (NCT02665065) is ongoing, with Iomab-B being studied in conjugation with a reduced-intensity conditioning regimen versus conventional care in patients with active, relapsed, or refractory AML. Iodine 131 does have limitations, as it emits gamma particles, which requires patients to remain in radiation isolation. This has led to the development of other anti-CD45 conjugated radionuclides, such as yttrium-90, which emit beta particles and thus do not require lead isolation. A Y90 conjugated-rat-derived anti-CD 45 Ig G2a monoclonal antibody, YAML568, showed good tolerability in a phase 1 study [122], and there is an ongoing phase 1 study of 90Y-DOTA BC8, an anti-CD45 antibody followed by allogenic stem cell transplantation for high-risk AML or MDS. (NCT01300572).

CD30 is a member of the tumor necrosis factor receptor superfamily. In addition to being expressed in other hematologic malignancies, CD30 is present on the surface of myeloblasts in patients with MDS and AML [123]. Brentuximab Vedotin (BV) is a chimeric anti-CD30 antibody conjugated to microtubule polymerization monomethylauristatin e (mmae) [124]. BV is FDA-approved in Hodgkin lymphoma and T-cell lymphoma, and is currently being evaluated in a phase I trial with re-induction chemotherapy in AML patients (NCT01830777), and as a single agent in phase 2 trial in AML and MDS patients (NCT01461538). A trial of BV with azacitadine was terminated due to slow patient recruitment (NCT02096042).

## 7. Cellular and Vaccine Therapies for MDS and AML

### 7.1. CAR T-Cell Therapies for MDS and AML 

Chimeric antigen receptor (CAR) T-cells are patient-derived T-cells transduced with viral vectors which express a chimeric antigen receptor for specific recognition of a tumor antigen, independent of major histocompatibility complex (MHC). CAR T-cells are composed of a single variable chain of an antibody domain linked to the signal transduction domain of the CD3 T-cell receptor (Figure 3). CAR T-cells have been successfully used to treat B-cell lymphomas; however, translation to use in AML has been slow to progress due to the scarcity of AML target antigens. Many potential targets such as CD123 and folate-receptor β are expressed in healthy myeloid precursors and T-cells [125]. The identification of a ligand specific to AML would eliminate the need for lymphodepletion therapy and its accompanying risks [126].

The use of natural killer group 2D (NKG2D) CAR T-cells is promising due to the significance of NKG2D as a highly conserved ligand upregulated in the setting of malignant transformation in AML [127]. In 2018, a phase 1 dose-escalation trial was conducted to evaluate the safety of a single infusion of a CAR T-cell therapy targeting the NKG2D ligand in 12 patients with AML, MDS, or R/R multiple myeloma without prior lymphodepleting conditioning (Table 4) [128]. In this trial, lymphodepleting therapy was eliminated to limit the potential for NKG2D ligand upregulation in healthy tissue. No dose-limiting toxicities, cytokine storm, or other adverse effects were noted specific to the NKG2D CAR T-cells. However, the response was limited to transient hematologic improvements at the highest dose, with disease stability noted in several subjects [128]. In vitro analysis revealed a robust response to NKG2D CAR T-cell therapy was triggered despite low density. Additionally, healthy peripheral blood mononuclear cells (PBMCs) were unaffected when pharmacologic NKG2D ligand induction was combined with histone deacetylase (HDAC) inhibition with valproic acid [128].

CD123, while not specific to MDS, has been shown as a marker of disease progression due to upregulation during MDS pathogenesis. CD123 is also a potential immune cellular therapy target for AML, but the non-specific nature may result in off-target effects, which are especially dangerous due to expression in vascular endothelium, associated with the risk of capillary leak [132]. A high rate of off-target toxicity has also been observed with tagraxofusp, an ADC targeting CD123 [119]. An in vitro and patient-derived MDS xenograft model demonstrated the efficacy of CD123-directed CAR T-cells as proof of concept, although this strategy has not yet been applied in clinical trials [133]. The CD123 CAR included a CD123 scFv, a CD28 costimulatory domain, a CD3ζ signaling domain, and a truncated EGFR domain. The CD123 CAR T-cells were tested against an MDS cell line, which was effectively eliminated after co-culture for 48 h. In the second phase of the study, CD123 CAR T-cells were co-cultured with specimens from four patients with high-risk MDS, with 50–100% elimination of primary MDS stem cells, with sparing of normal hematopoietic and progenitor cells [133]. 

Recently, FLT3-targeted CAR T-cells have been developed, including FLT3-CAR-R2, a construct modified with a rituximab-responsive mimotope off-switch between the hinge and scFv to mediate depletion [134]. This treatment was manufactured ‘off the shelf’ from healthy donor T-cells to eliminate the typical weeks-long wait time for an autologous T-cell product, which may be too long in rapidly progressing AML. The manufactured CAR T-cells were co-cultured with AML target cell lines to evaluate activity and longevity. In the in vitro model, CAR T-cell depletion with rituximab did not compromise disease remission, and FLT3 CAR R2 T-cells eliminated >80% of the AML blasts over a period of 48 h. The use of the rituximab off-switch allowed for post-treatment eradiation to limit hematopoietic toxicity of the FLT3 CAR R2 [134].

Another promising target for AML lies with the investigation of C-type lectin-like molecule-1 (CLL1) as a target highly expressed on AML stem cells and blasts but not on healthy HSCs. The first human use of anti-CLL1 CAR T-cells in a pediatric patient with relapsed AML was reported in 2020 [135]. The patient received an infusion of anti-CLL1 CAR T-cells after lymphodepleting chemotherapy, with adverse effects including transient hypotension and grade I-II cytokine release syndrome. The patient achieved complete remission, which was maintained nine months post-therapy, at the time of the paper submission [135].

Finally, a recent study investigated the use of CAR T-cells targeting CD70, which is expressed on most leukemic blasts with absent or low-level expression in normal marrow [136]. The investigators evaluated a panel of CD70-CARs, of which two constructs demonstrated both in vitro and in vivo efficacy for the elimination of AML cells without off-target toxicity to normal HSCs. The CD70 receptor CD27 CAR construct performed better than the scFv-based CAR [136]. These studies, while preliminary, suggest a potential benefit of CAR T-cell therapy for MDS and AML.

### 7.2. NK Cell Therapies for AML and MDS

Natural killer (NK) cells are type 1 ILCs that function as mediators of immunosurveillance and the elimination of malignant cells using activating and inhibiting receptors [137]. Once activated, NK cells release cytotoxic granules to lyse malignant cells and produce and release cytokines such as interferon gamma (IFN-γ) and tumor necrosis factor alpha (TNF-α) to moderate the adaptive immune response [132]. In patients with AML and MDS, NK cells are often dysfunctional and reduced in number [138], which poses a challenge in developing effective NK cell therapies. Therefore, therapeutic strategies to overcome NK cell dysfunction, restore NK immune surveillance, and enhance their function are under active investigation, including the use of tyrosine kinase inhibitors, hypomethylating agents, cytokines such as IL-2, and other immune-modulating agents [50,138]. 

NK cell-based therapy is generally well-tolerated and has the potential to induce complete remission in patients with AML. NK cell therapy is based on early studies involving the infusion of NK cells in conjunction with hematopoietic stem cell transplants, where it was shown to have a protective effect against AML relapse [139]. Challenges arise with promoting the persistence of NK cells with undesired activation of regulatory T-cells. In a recent study, 16 patients with R/R MDS or AML were treated with fludarabine and cyclophosphamide + total irradiation + adoptive immunotherapy with IL-2 activated haploidentical NK cells [129]. The therapy was well-tolerated, with three patients disease-free for greater than three years post-treatment, five patients bridged to allogeneic HSCT, and six with an objective response, indicating that high-risk MDS is responsive to NK cell therapy [129].

### 7.3. Vaccine Therapies for AML and MDS

Vaccine-based therapy is a rapidly evolving field of cancer immunotherapy designed to manipulate the immune system to recognize and eliminate tumor-specific cells through T-cell activation. Two common antigens include tumor-associated antigens (TAAs) and tumor-specific antigens (TSAs). TSAs are tumor- and patient-specific, so less likely to evoke autoimmunity compared to TAAs, which are self-antigens that are often over-expressed by tumor cells [140]. The Wilms tumor 1 (WT1) protein is an immunogenic tumor-associated antigen that is over-expressed in CD34 + MDS stem cells and AML blasts which elicits T-cell mediated myelosuppression and is a candidate antigen for vaccine therapy [130,141]. A recent pilot trial investigated the use of a WT1-specific T-cell receptor in patients with AML and high-risk MDS. Eight patients received WT1-specific TCR gene T-cell transfer without adverse events and were shown to maintain WT1 directed T-cells in peripheral blood for eight weeks and the capacity to mount an immune response to WT1 [130]. Two patients exhibited a reduction in bone marrow blasts transiently with recovered hematopoiesis, and five patients with persistent WT1 T-cells achieved more than 12-month survival (Table 4) [130].

NY-ESO-1 is an antigen expressed in various tumors, which elicits robust humoral and cellular immune responses and is another candidate TAA under investigation in vaccine trials for MDS and AML. A recent pilot study employed an HLA unrestricted NY-ESO-1 vaccine combined with decitabine in a study of nine high-risk MDS patients to induce an antigen-specific cytotoxic response against the malignant myeloid compartment in MDS [131]. Among seven patients that reached study completion, all patients demonstrated NY-ESO-I gene expression, and NY-ESO-1-specific CD4+ and CD8+ T-cell responses were observed in six and four patients, respectively (Table 4). Furthermore, NY-ESO-1 expressing myeloid cells were demonstrated to elicit a cytotoxic response from autologous NY-ESO-1-specific patient T-cells [131]. Further investigations combining NY-ESO-1 directed immune responses with hypomethylating agents are ongoing [142].

## 8. Conclusions

There has been remarkable progress over the past decade in understanding mechanisms of immune dysregulation in the pathogenesis of MDS and AML, as well as a refinement in knowledge of how immune effector cells exert their actions against tumor antigen targets. It is now well recognized that both innate and adaptive immunity are perturbed in the MDS and AML bone marrow microenvironment, and a variety of immune targets have emerged. A considerable advantage of immune therapies over cytotoxic chemotherapy is the potential for sustained immune surveillance and ongoing effector cell activity, which can lead to durable responses to treatment. IST in the form of ATG with cyclosporine continues to have a role in the treatment of selected patients with MDS, particularly patients with low and intermediate-1 disease and those with a hypocellular bone marrow. Immune checkpoint inhibitors administered as single agents have modest clinical activity in MDS and AML. However, upregulation of PD-L1 has been shown to induce resistance to AML therapy [129,143], and combination of BiTEs and other mAb constructs with PD-L1 blockade is a potential strategy to further improve therapeutic effector T-cell activity. The use of additional checkpoint inhibitors such as TIM-3 may also overcome T-cell exhaustion and rescue AML that is resistant to initial PD1/PD-L1 blockade [86]. Immune therapies have the potential to overcome chemotherapy-resistant forms of AML, including those with P53 mutations [144]. A variety of novel immune-modulating mAbs, engineered mAb constructs, ADC, and cellular and vaccine therapies have emerged as promising therapies for patients with R/R MDS and AML, with new therapeutic strategies being identified at an exponential pace. Although many clinicians do not have immediate access to experimental immunotherapies at their institutions, the vast array of immune therapy options underscores the importance of referring patients who have failed standard therapies for MDS and AML to centers that offer immune therapies on clinical trials. While patients with high-risk MDS and AML currently have a poor prognosis, the rapidly expanding number of novel immune therapies under investigation indicates a promising future with a plethora of more effective therapies to come.

## Figures and Tables

**Figure 3 cancers-13-05026-f003:**
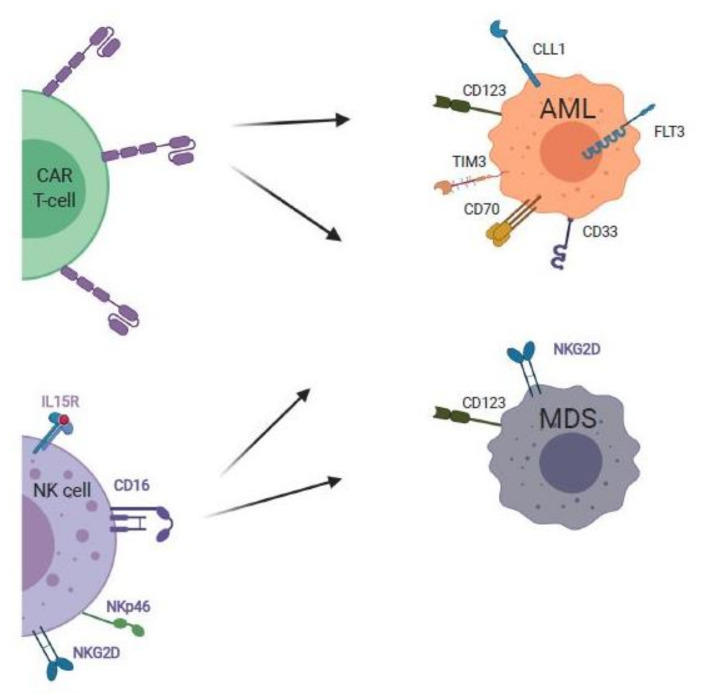
Adoptive cellular therapies for MDS and AML. Newer cellular therapies include chimeric antigen receptor T-cell (CAR T-cells) directed against natural killer group 2D (NKG2D), CD123, FLT3, and C-type lectin-line molecule-1 (CLL1), and CD70 expressed on MDS and AML cells. Another recent approach is to employ adoptive transfer of haploidentical natural killer (NK) cells to target and eradicate MDS and AML cells. Image created with BioRender.com.

**Table 1 cancers-13-05026-t001:** Clinical trials studying monoclonal antibodies as immune therapy in MDS and AML.

Study	Trial Identification	Mechanism	Study Population	Study Status	Results	MRD Status	Adverse Effects	Ref
Phase 1 non-randomized, open label, magrolimab and azacitidine in MDS/AML	NCT03248479	Anti-CD47	R/R AML or unfit treatment naïve AML or high risk MDS	Recruiting	High-risk MDS: 30/39 (91%) ORR with 42% CR, 24% marrow CR. Treatment-naïve AML: 22/34 (65%) ORR, 15 (44%) had CR. 15/21 TP53 mutants (71%) ORR, 10/21 (48%) had CR.	High-risk MDS: 22% patients with CR or CRi or marrow CR were MRD negative Treatment-naïve AML: N/A	Anemia, fatigue neutropenia, thrombocytopenia, infusion reaction, increased serum bilirubin, nausea	[66]
Phase 1 dose escalation trial, open label, single arm, magrolimab in hematologic malignancies	NCT02678338	Anti-CD47	R/R AML or high risk MDS	Completed	N/A	N/A	Decline in hemoglobin, increase in transfusion requirements, RBC agglutination with hemolysis. 9/19 (47%) with positive antibody screen	[69]
Phase 1 dose escalation trial, open label, CC-90002 in AML and high risk MDS	NCT02641002	Anti-CD47	R/R AML or high risk MDS	Terminated	Only 2/24 patients had stable disease, no ORR noted. No improvement in RBC transfusion requirements	N/A	Disseminated intravascular coagulation in 4 patients, diarrhea, thrombocytopenia, febrile neutropenia, anemia, increased AST/ALT, cough	[70]
Phase 1 dose escalation and expansion trial, open label, non randomized, TTI-621 in patients with hematologic malignancies and selected solid tumors	NCT02663518	Binds CD47	Relapsed or refractory hematologic malignancies, selected solid tumors	Recruiting	N/A	N/A	N/A	N/A
Phase 1/ 2 study, single arm, open label, ALX148 with venetoclax and azacitidine	NCT04755244	Fusion protein that blocks CD47-SIRPalpha pathway	R/R AML or unfit treatment naïve AML	Recruiting	N/A	N/A	N/A	N/A
Phase 1, dose escalation trial, single arm, open label, ipilimumab or nivolumab in relapsed hematologic malignancies after allogeneic HSCT	NCT01822509	Anti-CTLA4 monoclonal antibody	Variety of R/R hematologic malignancies	Active, not recruiting	MDS/AML specific results: CR in 4 extramedullary AML and 1 secondary AML patient out of 12 AML patients	N/A	Immune-mediated side effects in 6/28 patients (21%), causing 1 death, GVHD in 14% patients	[71]
Phase 2, non-randomized, open label, nivolumab and azacitidine with/without ipilimumab	NCT02397720	Anti-PD-1 monoclonal antibody	R/R AML, unfit treatment naïve AML	Recruiting	Nivolumab and azacitidine (n = 70): ORR 33% including 22% CR/CRi, 1 PR, 7 with hematologic improvement. ORR 58% in HMA naïve, 22% in HMA pre-treated patients. Nivolumab, azacitidine and ipilimumab (n = 36%): 19% with CR/Cri, 3% with PR, 14% with durable SD.	N/A	Grade 3 or 4 immune-mediated side effects in 11% patients Grade 3 or 4 immune-mediated toxicities in 19% patients	[72]
Phase 2, single arm, open label, PD-1 inhibition (nivolumab) in AML at high risk of relapse	NCT02532231	Anti-PD-1 monoclonal antibody	AML in remission with high risk of relapse	Recruiting	6 month RFS 57.1%, median RFS 8.48 months	7/9 (78%) with MRD positive stayed MRD positive and progressed. 2/9 cleared MRD. 1/6 with MRD negative status experienced recurrence	Grade 3 or 4 immune mediated toxicities were seen in 27% patients	[73]
Phase 1, non-randomized, open label, pembrolizumab and decitabine in newly diagnosed or R/R AML or MDS	NCT03969446	Anti-PD-1 monoclonal antibody	R/R AML, MDS	Recruiting	N/A	N/A	N/A	N/A
Phase 2, randomized, azacitidine and venetoclax with or without pembrolizumab in unfit AML patients	NCT04284787	Anti-PD-1 monoclonal antibody	Unfit patients with AML	Recruiting	N/A	N/A	N/A	N/A
Phase 2, randomized, intensive chemotherapy with or without pembrolizumab in fit AML patients	NCT04214249	Anti-PD-1 monoclonal antibody	Fit patients with AML	Recruiting	N/A	N/A	N/A	N/A
Phase 1b, single arm, pembrolizumab for graft vs leukemia effect in acute leukemia patients with relapse post allo-HSCT	NCT03286114	Anti-PD-1 monoclonal antibody	AML, ALL, or MDS in relapse after allo-HSCT	Recruiting	N/A	N/A	N/A	N/A
Phase 1b study, single arm, open label, pembrolizumab and entinostat in MDS after HMA failure	NCT02936752	Anti-PD-1 monoclonal antibody	MDS regardless of risk category or oligoblastic AML after HMA failure	Active, not recruiting	N/A	N/A	N/A	N/A
Phase 1, non-randomized, sabatolimab with HMAs in AML and high risk MDS	NCT03066648	Anti-TIM-3 monoclonal antibody	AML and high risk MDS	Active, not recruiting	AML (n = 34): ORR 41.2%, 12 month PFS was 44% High-risk MDS (n = 35): ORR 62.9%, 12 month PFS 58.1%	N/A	Thrombocytopenia, neutropenia, anemia, pneumonia, 7 immune mediate adverse events	[74]

Abbreviations: R/R, relapsed/refractory; ORR, objective response rate; CR, complete response; CRi, incomplete count recovery; MRD, minimal residual disease; HSCT, hematopoietic stem cell transplantation; ALL, acute lymphoblastic leukemia; HMA, hypomethylating agent; PFS, progression-free survival.

**Table 3 cancers-13-05026-t003:** Clinical trials of various Antibody Drug Conjugates in AML and MDS.

Study	Trial Identification	Mechanism	Study Population	Study Status	Results	MRD Status	Adverse Effects	Ref
Phase 1 trial of vadastuximab talirine monotherapy (VT) with HMA in patients with CD 33-positive AML patients.	NCT01902329	Anti-CD33	AML-CR with initial induction/consolidation or R/R-AML or declined treatment with high dose induction/consolidation	Completed	The composite remission rate (CR + CRi) was 70%. Median PFS and OS were 7.7 and 11.3 months, respectively	51% were MRD negative	Increased hematological toxicity	[105]
Phase 3 Study of VT versus placebo in combination with azacitidine or decitabine in the treatment of older patients with newly diagnosed AML	NCT02785900	Anti-CD33	Newly diagnosed AML	Terminated	N/A	N/A	Higher rate of death, including fatal infections in VT versus control arm	N/A
Phase 1, open label study of IMGN779 in adult patient with R/R CD 33 positive AML	NCT02674763	Anti-CD33	R/R AML	Completed	11/27 patients (41%) who received IMGN779 had a >30% reduction in bone marrow blasts	N/A	Febrile neutropenia, epitaxis, nausea, diarrhea, fatigue, abdominal pain and hypokalemia. Grade 3+ adverse events, most frequent: febrile neutropenia, bactermia, pneumonia, and anemia	[106]
Dose-escalation safety and pharmacokinetic study of AVE9633	NCT00543972	Anti-CD33	R/R AML	Terminated	Terminated due to absence of evidence of clinical activity to toxic doses	N/A	N/A	N/A
Phase 1/2 multicenter, open label study of IMGN632 monotherapy sdminstered intravenously in patients with CD 123-positive hematological malignancies	NCT03386513	Anti-CD 123	CD 123-postiive AML and other CD123-positive hematological malignancies	Recruiting	Four (33%) achieved an OR including one CR and three CRi	N/A	Decreased appetite, diarrhea, nausea, febrile neturopenia, peripheral edema, hypotension, sinus tachycardia	[107]
Phase 1, dose escalation and dose expansion trial of SGN-CD123A to evaluate safety, tolerability, and anti-tumor efficacy	NCT02848248	Anti-CD 123	R/R AML	Terminated	NA	N/A	Terminated due to safety concern	N/A
A Phase 1/2 study of SL-401 as consolidation therapy for adults with adverse risk AML in first CR and/or evidence of MRD in first CR	NCT02270463	Anti-CD 123	Adverse risk AML	Completed	N/A	N/A	N/A	N/A
Phase 1 trial of SL-401 in combination with azacitidine or azacitidine/venetoclax in AML, high-risk MDS or blastic plasmacytoid dendritic cell neoplasm	NCT03113643	Anti-CD123	AML, high-risk MDS, blastic plasmacytoid dendritic neoplasm	Recruiting	N/A	N/A	N/A	N/A
Phase 1 study combining escalating doses of radiolabeled BC8 antibody with fludarabine, and low-dose total-body irradiation followed by donor stem cell transplant and immunosuppresion therapy in treating older patients with advanced AML or high-risk MDS	NCT00008177	Anti-CD45	AML/High Risk MDS	Recruiting	CR in all patients and engraftment by day 28 of transplant. Median OS and DFS among all 58 patients—199 days and 159 days, respectively, and among the 21 patients treated at the MTD, 206 and 189 days, respectively	N/A	Infusion toxicites, chills, nausea, vomiting, respiratory symptoms such as throat or chest tightness, and hypotension	[108]
Phase 3 study of I 131 monoclonal antibody prior to allogeneic HSCT versus conventional care in older subjects with active, R/R AML	NCT02665065	Anti-CD45	Older patients with R/R AML	Recruiting	N/A	N/A	N/A	N/A
Phase 1 trial of brentuximab vedotin with re-induction chemotherapy with relapsed, CD30-positive AML	NCT01830777	Anti-CD30	R/R AML	Completed	N/A	N/A	N/A	N/A
Phase 2, open-label study of brentuximab vedotin in patients with CD30-positive nonlymphomatous malignancies	NCT01461538	Anti-CD30	AML, ALL or MDS	Completed	N/A	N/A	N/A	N/A
Phase 1/2 study of weekly schedule of brentuximab vedotin alone and in combination with azacytidine in CD 30 positive R/R AML	NCT01830777	Anti-CD30	R/R AML	Terminated due to slow accrual	N/A	N/A	N/A	N/A

Abbreviations: CR, complete remission; Cri, complete remission with incomplete count recover; OR, objective response; OS, overall survival; DFS, disease-free survival; MTD, maximum tolerated dose; PFS, progression-free survival.

**Table 4 cancers-13-05026-t004:** Clinical trials of CAR T-cells, NK Cells, and vaccine-based therapies for MDS and AML.

Study	Trial Identification	Mechanism	Study Population	Study Status	Results	MRD Status	Adverse Effects	Ref
CAR T-Cells								
Phase 1 dose-escalation trial	NCT02203825	NKG2D CAR T-Cells	AML, MDS, or R/R MM without prior lymphodepleting conditioning	Completed	Robust efficacy not observed; response % not reported	N/A	No adverse events related to NKG2D CAR-T cells	[128]
NK Cells								
Phase 1/2 adoptive immunotherapy trial	EudraCT number 2011-003181-32	IL-2 activated haploidentical NK cells	R/R high-risk myeloid malignancies	Completed	Objective response: 6/16 (38%) in patients with MDS and AML	N/A	Transient, treatable grade 3–4 toxicities including chills and nausea in 2/6 patients	[129]
Vaccine-based therapy								
Phase 1 dose-escalation trial	UMIN000011519.	WT1-specific TCR-T cell transfer	R/R AML and high-risk MDS expressing WT1 antigen	Completed	WT1 specific TCR T-cells survived in AML and MDS and displayed reactivity to WT1; hematologic efficacy was not established.	N/A	No dose-limiting toxicities	[130]
Single-center, phase 1 study	NCT01834248	NY-ESO-1 vaccine administered with standard dose decitabine induced NY-ESO-1 expression in circulating blasts.	MDS or low blast count AML	Completed	7 patients reached the end of the study, 7/7 (100%) demonstrated induction of NY-ESO-1 expression, 6/7 patients (86%) and 4/7 patients (57%) demonstrated NY-ESO-1 specific CD4+ and CD8+ T-lymphocyte response	N/A	Related to decitabine: cytopenias, elevated liver enzymes, fatigue, edema, diarrhea. Majority of patients developed localized skin reaction to the vaccine.	[131]

Abbreviations: CAR-T-cell, chimeric antigen receptor T-cell; NK cell, natural killer cell; MDS, myelodysplastic syndrome; AML, acute myeloid leukemia; MRD, measurable residual disease; RR M/M, relapsed/refractory multiple myeloma.

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
