# Peer review of "Immune Therapies for Myelodysplastic Syndromes and Acute Myeloid Leukemia"

_cancers, 2021, doi:10.3390/cancers13195026_

Round 1

Reviewer 1 Report

S Kapoor et al, review in this paper the aspects  of Immune Modulating Therapies for MDS and AML.

Here some comments and suggestions.

Firstly; the paper is very well writed and a large part of scientific data are very well presented in synthetic and didactic  manner.
Most importants fields of this topic are update and easy to read with large and complete  bibliographique refferences integrated in the text.

The topic are quite very important in this era of most personalised target therapy more and more attractive and well integrated in clinical trials for MDS and AML.
The authors decribed very clear and comprehensive manner the fondamental bases of mechanistic immune interaction in niche hematopoiesis and gene mutations implicated in pathway  dysfonction in clonal hematopoiesis in MDS and LSC AML.
Inflamatory features and autoimmuny play an important rôle and clear highlited by the authors ,  the importance of diagnosis associated of myelodisplasia but also therapeutical interventions reviewed in previously clinical trials.
Most recently it was described VEXAS syndrome associated in MDS patients , caused by a somatic mutation in UBA1 gene in myeloid hematopoietic progenitors and myeloid drived inflamatory symptoms ( Grayson et al, Blood 1rst July 2021). I think  it could  be important to added some phrases concerning this new entity mostly  associated  with MDS , with specificity at diagnosis morphology, clinical related to inflamatory symptoms  and proposal of therapy intervention decribed recently in some retrospective series or patients (Bourbon et al, Blood 2021;137(26):3682-3684. ...etc.

Finaly, this paper is an interesting review  for younger research and clinicians and pathologists and highlited complexity of phisiopathology in clonal MDS and AML hematopoiesis in some patients and connection between immune system comprehensive mechanistic dysruption to better guide therapy options associated to most specific drugs used in this type of diseases. 

Author Response

  1. In response to the comment from reviewer 1 we included a paragraph about VEXAS syndrome with three relevant references in lines 94-102.

Reviewer 2 Report

In the manuscript, the authors provide an interesting insight into the field of therapies involving immune agents on MDS and AML patients. Generally, this review appears to be carefully written, with an adequate usage of tables and figures to resume and simplify the relevant information presented in the main text. Nonetheless, there are some minor issues that I'd like to see adressed in a future version of the manuscript in order to improve this work.

  1. Firstly, this review results very descriptive, with authors reporting other study results mostly without any comments. Authors should include their hypothesis, for example on why these strategies have thus far failed to report strong results and on how to improve this kind of therapies
  2. The "Immune dysregulation in MDS and AML" section appear unbalanced, as only a few lines are dedicated to AML. Authors should correct this by adding other relevant results in AML.
  3. Strictly speaking, the antibody-drug conjugates are not immunomodulatory (unless the conjugated drug is an immunomodulant). Authors should explain the reasons behind the choice of including this strategy.
  4. Many of these strategies attempt to target NK cells. Authors report an impairment on the NK compartment. These two statements are in contrast. Please provide your view on this.
  5. Figure 2 should include more details on how these agents are constructed and immune cells targeted, to readily understand what is well explained in the text.
  6. Why issues involving the targeting of CD123 are reported only near the end, in the CAR-T section? Does the terapy with Fc-optimized antibodies suffer from the same limitations or not?

Other minor issues:

  • line 15 - there is an invertion of the verb (has there been progress)
  • line 24 - the past participle should be "selected". This typo occurs multiple times later in the text
  • line 43 - "cell lines" should be corrected in "cell lineages"
  • line 45 - there is a strange "blasts-2" that should be fixed
  • in the "Anti-PD-1 and anti-TIM-3 therapies" section authors have not provided a description for the presented agents. Are there any differences between the various antibodies?
  • line 609/610 - Iodine 131 does not cause gamma particles, but emits them. 

Author Response

  1. In response to reviewer 2, we added more commentary about the use of immune therapies and on why early studies with immune therapies have failed to show robust results in lines 195-197, 405-408, 429-430, 519-520, 585, 645-646, 763-767, 823-826, 829-832.
  2. In response to reviewer 2, we expanded the section on immune dysregulation in AML with relevant references in lines 186-196, 208-209.
  3. In response to reviewers 2 and 4, we changed the title from “Immune Modulating Therapies…” to “Immune Therapies” to reflect a more general category of mechanisms of action of these treatment approaches, including ADCs and cellular therapies.
  4. In response to reviewer 2, we included commentary on the challenge of NK cell dysfunction: “In patients with AML and MDS, NK cells are often dysfunctional and reduced in number [137], which poses a challenge in developing effective NK cell therapies. Therefore, therapeutic strategies to overcome NK cell dysfunction, restore NK immune surveillance, and enhance their function are under active investigation, including the use of tyrosine kinase inhibitors, hypomethylating agents, cytokines such as IL-2, and other immune-modulating agents [50, 137].” in lines 763-768.
  5. In response to reviewer 2, we edited the legend to figure 2 to include more detailed information on how the novel molecules are constructed in lines 305-317.
  6. In response to reviewer 2, we included sentences addressing the toxicity of anti-CD123 directed immune therapies. There are not mature studies detailing this for the CD123 Fc-optimized mAb. We indicate this on lines 429-431. We also indicate this toxicity associated with CD123 directed ADCs on lines 723-724.
  7. We corrected the typographical errors and grammar throughout the manuscript as suggested by reviewers 2, 3. They are indicated in the tracked version of the manuscript.

Reviewer 3 Report

Sargam Kapoor et al. present a quality and well-written review manuscript describing immune modulating therapies for myelodysplastic syndromes and acute myeloid leukemia.

Authors suggest that it is well recognized that there is an immune component to the pathogenesis of MDS and AML, but until recently, immune modulatory therapies have played a limited role in the treatment of these diseases. Insights into the immunobiology of MDS and AML have yielded emerging therapies, including novel monoclonal antibodies engineered to enhance antibody-dependent cytotoxicity and disrupt immune checkpoint interactions, bispecific and trispecific T-cell engaging antibodies, antibody drug conjugates, vaccine therapies, and cellular therapeutics including chimeric antigen receptor T-cells and NK cells. These emerging immune therapies have the potential to transform current treatment paradigms.

Authors provide an update on current knowledge of immune dysregulation in MDS and AML, and overview novel immune modulating therapies for these life threatening bone marrow diseases, with particular emphasis on emerging treatment approaches.

Authors describe immune dysregulation in MDS and AML, as well as immunosuppression trials for MDS. A substantial emphasis is made on monoclonal antibody therapy for MDS and AML (anti-CD47), immune checkpoint inhibitor therapies, antibody dependent cellular cytotoxicity. In addition, authors also report recent advances in the field of Bispecific T-cell engagers, dual-affinity re-targeting molecules, bi- and tri-specific killer cell engager, antibody drug conjugate, cellular and vaccine therapies for MDS and AML.

Finally, authors conclude that a variety of novel immune modulating mAbs, engineered mAb constructs, ADC, and cellular and vaccine therapies have emerged as promising therapies for patients with R/R MDS and AML, with new therapeutic strategies being identified at an exponential pace. While patients with high risk MDS and AML currently have a poor prognosis, the rapidly expanding number of novel immune modulating therapies under investigation indicates a promising future with a plethora of more effective immune therapies to come.

Other comments:

1) Please check for typos throughout the manuscript.

2) In the paragraph that describes p53 mutant tutors (lines 257-274) - authors are kindly encouraged to cite the following article that describes various aspects of humoral and cell based immunotherapy of p53 mutant tumors. DOI: 10.3389/fimmu.2021.707734

Overall, the manuscript is highly valuable for the scientific community and should be accepted for publication.

Author Response

  1. We corrected the typographical errors and grammar throughout the manuscript as suggested by reviewers 2, 3. They are indicated in the tracked version of the manuscript.

  1. In response to reviewer 3 we included the suggested reference in the conclusion on lines 825-826.

Reviewer 4 Report

The authors present a review about MDS and AML. The background to this are numerous new clinical studies that attempt to improve the prognosis of patients via immunological intervention.
The weakest point of the manuscript is the simple summary: in terms of content, the statements on lines 14 to 16 duplicate each other, but it is not clear that immunological therapies are summarized here in a very focused way in their full breadth.
At the introduction, MDS and AML are briefly introduced. For the reader less familiar with classifications, a direct link to the WHO classification would be helpful. 
Immune dysregulation is well described and visualized in Figure 1. 
Almost historical is the presentation of immunosuppression with ATG, but certainly useful in the context of a review. Therapy with monoclonal antibodies is well structured and neatly presented. Tables and figures provide a good overview. The reviewer only has a problem with paragraph 4.3, because it is not about immunomodulation, but about targeted immunotherapy against tumor antigens. The same problem appears again later with 6.2/6.3 and the cellular therapies. This should be more clearly distinguished from immunomodulatory therapies. Bi-/trispecific antibodies and ADCs follow. This is followed by the cellular therapies, although clinically these are still very early for MDS and AML.  
Overall, it is a fascinating development.

Author Response

  1. In response to reviewers 2 and 4, we changed the title from “Immune Modulating Therapies…” to “Immune Therapies” to reflect a more general category of mechanisms of action of these treatment approaches, including ADCs and cellular therapies.
  2. In response to reviewer 4, we included a link to the WHO classification on lines 48-49.
  3. In response to reviewer 4, we rewrote the simple summary on lines 11-21.

Round 2

Reviewer 4 Report

The authors followed the reviewer's proposals.